# Combination Therapy of 0.1% Fluorometholone and 0.05% Azelastine in Eyes with Severe Allergic Conjunctival Diseases: A Randomized Controlled Trial

**DOI:** 10.3390/jcm11133877

**Published:** 2022-07-04

**Authors:** Minjie Chen, Bilian Ke, Jun Zou, Lan Gong, Yan Wang, Chaoran Zhang, Jianjiang Xu, Anji Wei, Jiaxu Hong

**Affiliations:** 1Department of Ophthalmology, Eye and Ear, Nose, Throat Hospital of Fudan University, 83 Fenyang Road, Shanghai 200031, China; minjie.chen@fdeent.org (M.C.); lan.gong@fdeent.org (L.G.); yan.wang@fdeent.org (Y.W.); chaoran.zhang@fdeent.org (C.Z.); jianjiang.xu@fdeent.org (J.X.); 2Key Laboratory of Visual Impairment and Restoration of Shanghai, Fudan University, 83 Fenyang Road, Shanghai 200031, China; 3Key Myopia Laboratory of National Health Commission of China, 83 Fenyang Road, Shanghai 200031, China; 4Key Laboratory of Myopia, Chinese Academy of Medical Science, 83 Fenyang Road, Shanghai 200031, China; 5Department of Ophthalmology, Shanghai General Hospital, Shanghai Jiaotong University School of Medicine, Shanghai 200080, China; kebilian@126.com; 6Department of Ophthalmology, Shanghai Tenth People’s Hospital, Tongji University School of Medicine, Shanghai 200072, China; zoujun70@126.com; 7Department of Ophthalmology, The Affiliated Hospital of Guizhou Medical University, 28 Guiyi Road, Guiyang 550031, China

**Keywords:** atopic keratoconjunctivitis, vernal keratoconjunctivitis, giant papillary conjunctivitis, fluorometholone, azelastine

## Abstract

This study sought to evaluate the efficacy of the isolated use of fluorometholone compared with the combined use of azelastine and fluorometholone for the treatment of severe allergic conjunctival disease (ACD). One hundred and eleven patients with severe ACD were randomized into two groups: one treated with topical 0.1% fluorometholone combined with 0.05% azelastine and the other with 0.1% fluorometholone alone. The Ocular Surface Disease Index (OSDI) and the signs of keratopathy, palpebral conjunctiva papillae and conjunctival congestion were scored before and at one, two and six weeks after treatment and compared between the groups. The intra-ocular pressure (IOP) was also monitored. There were no significant differences between the groups in the baseline mean scores of signs and OSDI scores, which gradually improved at all visits after therapy in both groups. Although the time effect was significant for all the parameters (all *p* < 0.001), the reduction in corneal involvement scores from week 2 to week 6 was insignificant in both groups (*p* = 0.460 for the steroids group and *p* = 0.074 for the combination group). All signs and symptoms were significantly more improved in the combination group than in the isolated group at each control visit. IOP remained stable at all visits (all *p* < 0.001), except one patient in each group had elevated IOP over 21 mmHg. While both the isolated use of fluorometholone and combined use of azelastine and fluorometholone are effective in alleviating the signs and symptoms of severe ACD, optimal response can be achieved with adjunctive treatment including azelastine.

## 1. Introduction

Allergic conjunctivitis disease (ACD) affects over 20% of the population and has a significant negative impact on ocular health, quality of life and productivity [1,2,3]. ACD is broadly divided into five types: seasonal allergic conjunctivitis (SAC), perennial allergic conjunctivitis (PAC), atopic keratoconjunctivitis (AKC), vernal keratoconjunctivitis (VKC) and giant papillary conjunctivitis (GPC). Of these, AKC, VKC and GPC are categorized as more complex pathologies as well as severe types of ACD due to the corneal involvement potentially [4,5]. In addition to ocular redness, itching and tearing, severe types of ACD are often accompanied by keratopathies, such as a shield ulcer or corneal plaque, corneal scarring and limbal stem cell deficiency, leading to visual morbidity if not treated promptly [4,5,6]. 

Most cases of ACD should be managed with pharmacological intervention. Dual-acting agents that act by blocking histamine receptor and stabilizing mast cells are proven to be safe and effective in relieving the signs and symptoms of patients with ACD [2,4,6,7]. However, patients with severe ACD often require stronger anti-inflammatory agents, such as topical corticosteroids and immunomodulators. While topical cyclosporine and tacrolimus exhibit excellent performance in patients suffering from severe ACD [4,5,6,7,8,9,10], several limitations restrict their administration. First, these nonsteroidal immunomodulators can result in an intense stinging sensation, leading to poor compliance [5,11,12]. Second, tacrolimus is still an off-label drug and is not commercially available in our countries. Moreover, various authors have failed to reach an agreement with the dose and frequency [7]. Even though topical corticosteroids could be used in such cases, they are not recommended for long-term use due to adverse effects, such as steroid-induced glaucoma, cataract, delayed wound healing and increased susceptibility to infection or superinfection, especially in children [3,4,6,7]. This is because children are more prone than adults to increased intraocular pressure (IOP) following steroid therapy, and more than 60% of them are steroid responders [4,13]. There are two categories of steroids used in ACD according to the intraocular penetration. High-potency steroids, including prednisolone phosphate 1% and dexamethasone sodium phosphate 0.1%, should never be used as a first treatment choice due to their high incidence of side effects [7]. Low-potency steroids, such as fluorometholone acetate 0.1%, are recommended as first-line steroids for active VKC cases due to their low intraocular penetration [7,14]. Regular monitoring of IOP in cases with prolonged low-potency steroid use is critical.

A prospective, randomized study was conducted to compare the efficacy of topical fluorometholone alone or with adjuvant azelastine in severe ACD. The purpose of the study was to evaluate whether azelastine could be used as a supplementary agent, strengthening the effect of topical corticosteroids in this group of patients.

## 2. Materials and Methods

This randomized and prospective study was conducted between 1 April 2019, and 30 October 2019, in 3 clinical centers in China. The study was registered at www.chictr.org.cn (accessed on 28 March 2019) (clinical trial accession number: ChiCTR1900022169). The study protocol was presented and approved by the Institutional Review Board of the Eye and ENT Hospital of Fudan University and followed the tenets of the Declaration of Helsinki. Informed written consent was obtained from all patients or their parents before participation in the study. 

### 2.1. Patients

The study enrolled patients with VKC, AKC and GPC who visited the Eye and ENT Hospital, Shanghai General Hospital and Shanghai Tenth People’s Hospital. In addition to the similar symptoms of redness, tearing, itching, photophobia, foreign body sensation, and so on, VKC, AKC and GPC were diagnosed with different rules. The criteria used for GPC patients were contact lens users and was characterized by the presence of “giant” papillae on the superior tarsal conjunctiva [6]. VKC was mainly diagnosed based on palpebral conjunctiva papillae and corneal involvement. As the borderline for distinguishing between AKC and VKC is vague in actual clinical practice, AKC diagnoses were based on chronic keratoconjunctivitis associated with atopic dermatitis on the patient’s face [5,15,16]. The keratopathy included the presence of a shield ulcer, superficial keratitis and Horner–Trantas dots.

Exclusion criteria included a history of ocular surgery in the previous 3 months or any other ocular conditions, such as associated corneal diseases, uveitis, glaucoma and optic atrophy, that could affect the trial results. Individuals concomitantly using any steroids, immunosuppressives, anti-allergic ocular medication within the previous 4 weeks were also excluded from the study, and only physical measures were allowed during that period. 

Two-sample *t*-tests allowing unequal variance method is used to calculate the sample size. The type I error probability of α is set as 0.05. The power of test (1-β) is set as 0.90 with two-sided test. According to OSDI changes from baseline to 6 weeks after therapy, which are supposed to be 19.63 ± 5.56 and 23.7 ± 5.51, respectively, the sample size is calculated as 80 cases. Assuming that the loss of follow-up rate is 20%, at least 100 patients are needed to be included as research objects. 

Eligible patients were randomized (1:1) according to a computer-generated predetermined randomization list to 6 week treatment with fluorometholone 0.1% (Flumetholon, Santen, Osaka, Japan) alone or with adjuvant azelastine 0.05% (Azeptin, Tubilux Pharma S.p.A., Rome, Italy) (Figure 1). The fluorometholone 0.1% was administered three times daily for one week and then reduced to twice daily until the end. Topical azelastine was prescribed twice daily for a six-week period. Follow-ups were set at 1 week, 2 weeks and 6 weeks after initiation of treatment. Participants were asked to suspend contact lens use from the beginning of the study.

### 2.2. Outcome Measures

At the time of enrollment, usage of contact lens and atopic dermatitis were inquired and recorded. The slit-lamp photography was applied to thoroughly assess the ocular condition at all visits. Conjunctival hyperemia, corneal involvement and palpebral conjunctiva papillae were assessed using a 4-point scale at all visits, as described previously [8,17] (Table 1). The mean sign values were assigned by two expert ophthalmologists independently. The Ocular Surface Disease Index (OSDI) questionnaire, consisted of 12 questions involving the frequency of the experienced symptoms, their impact on vision-related QoL, and the presence of any environmental triggers was used at all visits [7,8]. If needed, guardians aided children in completing the OSDI. 

### 2.3. Safety

IOP measured with a non-contact tonometer (NCT), lens opacification or other possible adverse events were observed to assess the safety and side effects of the treatment at each visit. 

### 2.4. Statistical Analyses

The data were statistically analyzed using Stata 14.0 (Stata Corp., College Station, TX, USA). The eyes with the worst total signs values were chosen as the study eyes. The right eye was enrolled with identical sign scores in both eyes. To evaluate the time course difference, an analysis of variance (ANOVA) or Kruskal–Wallis test was performed along with the Bonferroni test. Between-group differences were checked with a paired t-test or matched-pairs signed-rank test. The χ^2^ tests were applied with numeration data to compare between-group difference. *p*-values of less than 0.05 were considered to be statistically significant.

## 3. Results

### 3.1. Baseline Characteristics

The frequency of lost to follow-up was comparable between the two groups (*p* = 0.775), with rates of 12.0% and 10.0% for the fluorometholone and combination groups, respectively. The remaining 100 patients completed the treatment and were analyzed thereafter (Figure 1). There were 31 children in the monotherapy group compared with 25 children in the combination group (*p* = 0.227). No significant between-group differences were observed in age, gender, duration, history of contact lens use and disease distribution (Table 2).

### 3.2. Improvements of Symptom Scores in Both Groups 

At baseline, the mean OSDI scores were 27.50 ± 8.46 and 25.76 ± 10.14 in the steroids and combination groups, respectively. There was no significant difference between groups (*p* = 0.3537). At the follow-up visits, an obvious decrease in OSDI scores was found in both groups (both *p* = 0.0001). The OSDI scores decreased from the baseline level to 3.74 ± 3.08 and 2.06 ± 2.19 in the steroids and combination groups, respectively. Although the time effect was significant in both groups (all *p* < 0.001), lower OSDI scores were seen in the combination group than in the steroids group at all follow-up visits (*p* = 0.0064, <0.0001 and 0.0009 at 1 week, 2 weeks and 6 weeks, respectively) (Figure 2).

### 3.3. Improvements of Sign Scores in Both Groups

There was no significant difference in the sign scores between the groups at baseline (all *p* > 0.2) (Table 3). Marked and progressive improvement with corneal involvement was noted within 2 weeks after the beginning treatment in both groups (all *p* < 0.001). However, the reduction in corneal involvement scores from week 2 to week 6 was insignificant in both groups (*p* = 0.460 for the steroids group and *p* = 0.074 for the combination group). Moreover, a significant synergistic effect on corneal involvement was observed with adjunctive topical 0.05% azelastine use at all visits after therapy (*p* = 0.0037, 0.0004 and 0.0069 at week 1, week 2 and week 6, respectively) (Figure 3).

At baseline, the mean palpebral conjunctiva papillae scores were 2.80 ± 0.45 and 2.76 ± 0.48 in the steroids and combination groups, respectively. There was no significant difference between groups (*p* = 0.6576). At the follow-up visits, a decrease in palpebral conjunctiva papillae scores was found in both groups (Table 3). The mean papillae scores decreased from the baseline level to 1.20 ± 0.64 and 0.78 ± 0.46 in the steroids and combination groups, respectively (from baseline to 6 weeks). The papillae scores decreased with treatment in both groups by 6 weeks, with a significant time effect (both *p* < 0.001). At all follow-ups after medication, there were fewer eyes with palpebral conjunctiva papillae in the combination group than in the isolated group (all *p* < 0.001) (Figure 3).

The mean conjunctival hyperemia scores at baseline were comparable between groups (3.01 ± 0.14 and 2.96 ± 0.35 in the monotherapy and combination groups, respectively; *p* = 0.2551). The mean conjunctival hyperemia scores were significantly reduced after 1 week of medication use in both groups, and the improvements continued until the end of the observation period (both *p* < 0.001) (Table 3). The absolute change in conjunctival hyperemia scores was much greater with the inclusion of topical azelastine at all three visits (*p* = 0.0128, < 0.0001, 0.0097 at week 1, week 2 and week 6, respectively) (Figure 3).

### 3.4. Safety

Baseline IOP was comparable in both groups (16.07 ± 1.71 and 16.17 ± 1.54 mmHg in the monotherapy and combination groups, respectively) (*p* = 0.7730). One patient in each group had elevated IOP over 21 mmHg two weeks after beginning medication. In these cases, the researcher discontinued therapy and switched to immunomodulators. The IOP of the two patients returned to normal levels in one week after the medication change. Changes in IOP from baseline to the 6-week follow-up were not statistically significant (16.28 ± 1.72 and 16.30 ± 1.52 mmHg in the monotherapy and combination groups, respectively) (*p* = 0.9364). None had lens opacity and cataract formation during the treatment course. Although we did not check the visual acuity (VA) in the current study, none of the patients complained about decreased VA. 

## 4. Discussion

While both topical fluorometholone and azelastine are effective in alleviating the signs and symptoms of ACD, their combined use is not common. In current study, ocular signs and OSDI scores were significantly improved after one week of treatment with fluorometholone alone as well as in combination with azelastine. The addition of azelastine significantly improved the therapeutic response, as evidenced by the lower sign and symptom scores at all visits after therapy. In contrast, improvement in ocular congestion was observed earlier in patients using fluorometholone along with olopatadine than patients using flurometholone monotherapy, however, comparable results were found after 8 weeks of therapy in GPC patients [18]. The different inclusion criteria may account for the inconsistence. Usually, patients with VKC and AKC had more complex ocular pathologies than those with GPC [4,5,6] according to the corneal involvement.

As a dual-acting agent, azelastine has the major advantage of quick action with immediate histamine receptor blocking and long-term stabilization of mast cell [7]. Topical steroids are the most effective anti-allergic drugs and are the treatment of choice in cases with persistent symptoms with keratopathy, congestion and giant papillae [7]. Moreover, the synergistic effect of azelastine was observed to cause more complete immunosuppression than fluorometholone alone in the current study. Due to the quick relief of itching and redness, combination therapy resulted in a greater reduction in OSDI scores one week after medication initiation and continued to produce superior results thereafter compared to the monotherapy group. A similar trend was observed in all the signs as well. Although topical fluorometholone alone was also effective in the treatment of severe ACD in this study, the earlier and quicker relief was important for the patients, especially for children. Indeed, intolerable symptoms of itching, photophobia, mucoid discharge, redness and foreign body sensation may be severe enough to incapacitate the subject, hindering the performance of daily activities [1,3,4,7]. Some patients are unable to stop rubbing their eyes, potentially leading to keratoconus and significant visual deterioration [7,19]. Thus, alleviating the signs and symptoms of ACD as soon as possible is of great value in clinical practice.

Furthermore, the time effect was significant for all the parameters except for the change of corneal involvement from week 2 to week 6. The presence of conjunctival congestion is a hint of clinical activity [7]. However, while conjunctival hyperemia and palpebral conjunctiva papillae were consistently improved from week 2 to week 6 in both groups, corneal involvement did not change significantly in the present study. In addition to mechanical abrasions with the papillae, the chemical mediators released by eosinophils are found to be cytotoxic, blocking the wound healing of the corneal epithelium [7,20]. This may account for delayed healing of the cornea at week 6 until the inflammation is resolved. In other words, severe ACD usually requires long-term intervention more than 6 weeks. 

With respect to the topical steroids, their safety and potential side effects are of greatest concern, especially increased IOP. Fluorometholone 0.1% is a less potent topical corticosteroid, but it has also been shown to lead to elevated IOP with much less risk [21,22,23]. In addition, children are more prone to steroid responsiveness compared with adults, and children with VKC have an even higher steroid response rate [22,24]. In our study, topical fluorometholone was halted for two children after two weeks due to elevated IOP > 21 mmHg. After discontinuing the medication, their IOP recovered to normal levels. This is consistent with a previous study reporting that the resolution of elevated IOP following steroid withdrawal in subjects using topical steroids for less than eight weeks is common [25]. The remainder of the subjects in both groups had stable IOP until the end of the study. According to published data, the level of steroid response is highest for topical dexamethasone eye drops [22,26,27]. Fluorometholone has poor ocular penetration because the metabolization of fluorometholone to dihydrofluorometholone during its passage through cornea into the aqueous humor avoids the risk of elevated IOP and steroid-induced glaucoma [28]. While topical steroids are easily available in China, general physicians and general ophthalmologists should be educated in the use of low-potent steroids and to monitor for side effects like elevated IOP, which could prevent blindness in young children. 

The current study has several limitations. First, visual acuity was not recorded in this study, which is important when interpreting the safety of the drugs. Second, the treatment course of 6 weeks may not be sufficient for severe ACD, such as VKC, and the prolonged use of topical steroids may cause more side effects. Third, we did not analyze efficacy according to the type of disease. There may be different therapeutical responses among VKC, AKC and GPC patients. Further research is needed on the treatment of VKC, AKC and GPC to test the repeatability of current results.

In conclusion, we explored the synergistic effect of fluorometholone and azelastine in severe cases of ACD. The results show that adjunctive use of azelastine can improve the therapeutic response of fluorometholone. Moreover, topical fluorometholone and azelastine are easily available. Thus, we recommend the combination of fluorometholone and azelastine to achieve optimal response in severe cases of ACD. However, it is important to monitor closely for side effects, including elevated IOP.

## Figures and Tables

**Figure 1 jcm-11-03877-f001:**
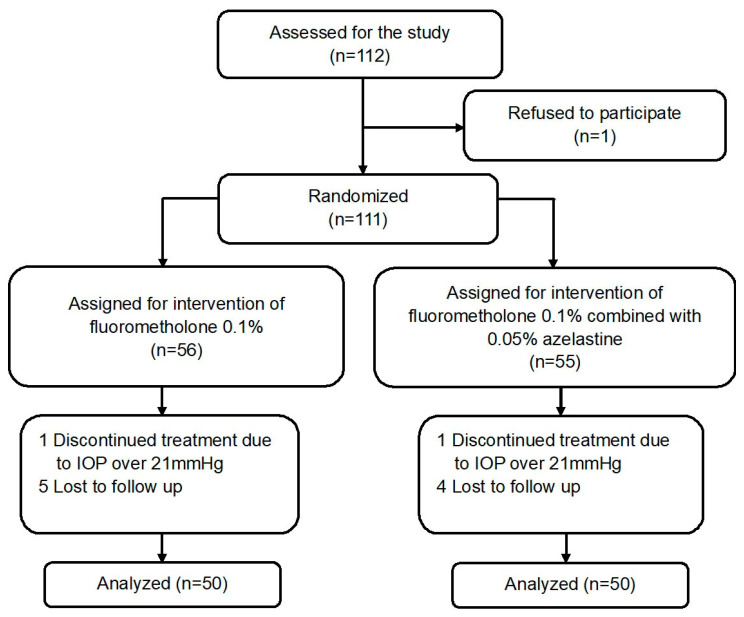
Participant flowchart.

**Figure 2 jcm-11-03877-f002:**
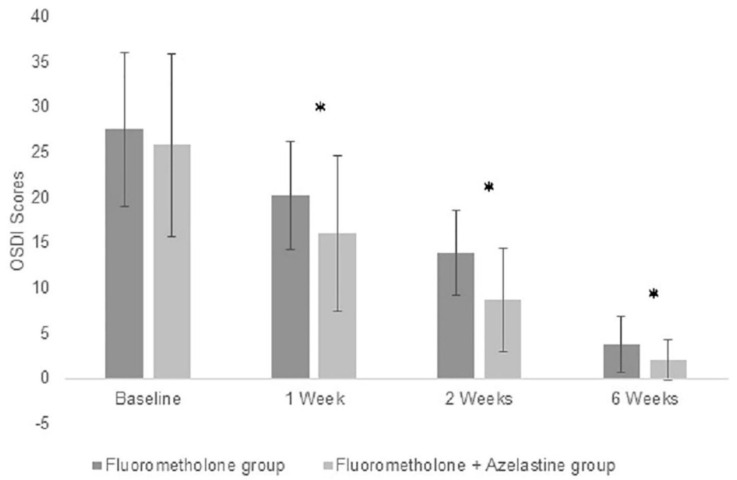
Improvement of OSDI scores with topical fluorometholone alone or combined with azelastine at baseline. * *p* < 0.05 compared between group. OSDI = Ocular Surface Disease Index.

**Figure 3 jcm-11-03877-f003:**
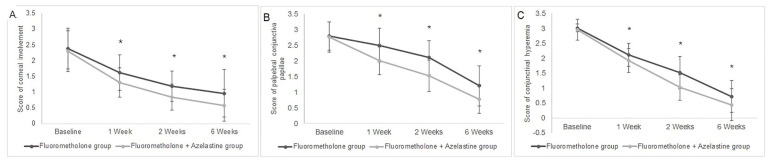
Improvement of corneal involvement, palpebral conjunctiva papillae and conjunctival hyperemia with 0.1% topical fluorometholone alone or combined with 0.05% azelastine from baseline. (**A**) The corneal involvement score was significantly decreased at all follow-ups in both groups (*p* < 0.0001). A significant synergetic effect was found with combined topical 0.05% azelastine at 1, 2 and 6 weeks from baseline (*p* = 0.0037, 0.0004 and 0.0069 at week 1, week 2 and week 6, respectively). (**B**,**C**) Combined treatment showed priority compared with isolated treatment at all visits in palpebral conjunctiva papillae and conjunctival hyperemia (all *p* < 0.02). * *p* < 0.05 compared between group.

**Table 1 jcm-11-03877-t001:** Grading scales for clinical signs.

Signs	Grading	Definition
Conjunctiva hyperaemia	0	None
	1	Dilatation of several vessels
	2	Dilatation of many vessels
	3	Diffuse dilated vessels in all the bulbar conjunctiva
Palpebral conjunctiva papillae	0	None
	1	Flat papillae
	2	Elevated papillae in <1/2 of the upper palpebral conjunctiva
	3	Elevated papillae in >1/2 or more of the upper palpebral conjunctiva or giant papillae (papillae size ≥1 mm)
Corneal involvement	0	None
	1	Superficial punctate keratitis
	2	Exfoliation superficial punctate keratitis
	3	Shield ulcer

**Table 2 jcm-11-03877-t002:** Patient demographic characteristics at baseline.

	Fluorometholone Group (*n* = 50 Eyes)	Fluorometholone + Azelastine Group (*n* = 50 Eyes)	*p*
Age (y)	17.68 ± 11.09	19.58 ± 11.74	0.4171
Male (%)	72.0%	64.0%	0.391
Duration of course (months)	47.56 ± 36.24	41.4 ± 34.79	0.3491
History of contact lens	25.0%	30.0%	0.248
Numbers of VKC/GPC/AKC (eyes)	31/10/9	27/16/7	0.385

AKC: atopic keratoconjunctivitis, VKC: vernal keratoconjunctivitis, GPC: giant papillary conjunctivitis.

**Table 3 jcm-11-03877-t003:** Clinical signs before and after therapy.

	Corneal Involvement			Palpebral Conjunctiva Papillae			Conjunctival Hyperemia		
	Fluorometholone Group	Fluorometholone + Azelastine Group	*p*	Fluorometholone Group	Fluorometholone + Azelastine Group	*p*	Fluorometholone Group	Fluorometholone + Azelastine Group	*p*
Baseline	2.38 ± 0.64	2.30 ± 0.65	0.5341	2.80 ± 0.45	2.76 ± 0.48	0.6576	3.01 ± 0.14	2.96 ± 0.35	0.2551
Week 1	1.62 ± 0.57	1.30 ± 0.46	0.0037	2.50 ± 0.54	2.00 ± 0.45	<0.0001	2.12 ± 0.39	1.92 ± 0.40	0.0128
Week 2	1.18 ± 0.48	0.84 ± 0.42	0.0004	2.1 ± 0.54	1.52 ± 0.50	<0.0001	1.52 ± 0.54	1.02 ± 0.43	<0.0001
Week 6	0.96 ± 0.75	0.58 ± 0.50	0.0069	1.2 ± 0.64	0.78 ± 0.46	0.0003	0.72 ± 0.54	0.44 ± 0.54	0.0097
P	0.0001	0.0001		0.0001	0.0001		0.0001	0.0001	

P indicates the within-group differences. *p* indicates the between-group differences.

## Data Availability

The data presented in this study are available on request from the corresponding author.

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
