# Peer review of "Combination Therapy of 0.1% Fluorometholone and 0.05% Azelastine in Eyes with Severe Allergic Conjunctival Diseases: A Randomized Controlled Trial"

_jcm, 2022, doi:10.3390/jcm11133877_

Round 1

Reviewer 1 Report

In most severe cases of allergic conjunctivitis, patients are either dependent on or resistant to steroid with and without anti-histamin therapy, which can have many side effects for the patient's eye health and vision. We inevitably have to use topical immunosuppressive drugs such as cyclosporine and tacrolimus in tis situations. Therefore, tacrolimus is still an off-label drug and is not commercially available in many countries. This study evaluates a low-potency steroid and antihistamine as an alternative treatment in these cases.

The authors’ definition of severe ocular allergy should be defined in all subcategories.

Please briefly explain the Ocular Surface Disease Index (OSDI) that used at all visits in the method section.

The authors only discussed about monitoring for side effects like elevated IOP and other steroids side effects such as lens opacity and cataract formation were not evaluated, and as mentioned in the limitation visual acuity was not recorded in this study, which is important when interpreting the safety of the drugs on the eye health.

Author Response

Dear reviewer:

We really appreciate for the reviewers’ comments on our manuscript titled “Combination Therapy of 0.1% Fluorometholone and 0.05% Azelastine in Eyes with Severe Allergic Conjunctival Diseases: A randomized controlled trial”, which are helpful to the revision of our manuscript, and it definitely improve the quality of our manuscript. In this article, we have revised the manuscript in accordance with the reviewers’ comments and carefully proofread the manuscript to correct typographical, grammatical, and bibliographical errors. The revised texts are in red. In addition, our responses to the reviewers’ comments one by one are listed below. We hope the revised manuscript will be more acceptable for publication in your journal.

Best regards

Yours Sincerely,

Jiaxu Hong

Jiaxu Hong MD, PhD, MPA

Email: jiaxu_hong@163.com

Address: Department of Ophthalmology and Visual Science, Eye, and ENT Hospital, Shanghai Medical College, Fudan University, 83 Fenyang Road, Shanghai, China; Department of Ophthalmology, The Affiliated Hospital of Guizhou Medical University, Guiyang, China, Telephone: +86-021-64377134 Fax: +86-021-64377151

In most severe cases of allergic conjunctivitis, patients are either dependent on or resistant to steroid with and without anti-histamine therapy, which can have many side effects for the patient's eye health and vision. We inevitably have to use topical immunosuppressive drugs such as cyclosporine and tacrolimus in tis situations. Therefore, tacrolimus is still an off-label drug and is not commercially available in many countries. This study evaluates a low-potency steroid and antihistamine as an alternative treatment in these cases.

The authors’ definition of severe ocular allergy should be defined in all subcategories.

Response: We added the definition of 3 subcategories in the method part of the revised paper.

Please briefly explain the Ocular Surface Disease Index (OSDI) that used at all visits in the method section.

Response: We added the introduction of OSDI in the method part of the revised paper.

The authors only discussed about monitoring for side effects like elevated IOP and other steroids side effects such as lens opacity and cataract formation were not evaluated, and as mentioned in the limitation visual acuity was not recorded in this study, which is important when interpreting the safety of the drugs on the eye health.

Response: None had lens opacity and cataract formation during the treatment course. Though we didn’t check the visual acuity (VA) in the current study, none of the patients complained about decreased VA at every visits. Of course, the lack of visual acuity record is a big limitation in our paper. We added the expression in the revised paper.

Reviewer 2 Report

- It was good to include data at 6 weeks as the mast cell stabilizing effects of azelastine take time to work. It would be good to have even longer data, utilizing the effects of azelastine.

- The conclusion makes sense as using azelastine in the compound takes advantage of the anti-histamine effect as well as the immunomodulatory effect of FML. 

- You could comment on FML's poor ocular penetration thus limiting steroid IOP response and keeping its effect to the corneal surface.

Author Response

Dear reviewer:

We really appreciate for the your comments on our manuscript titled “Combination Therapy of 0.1% Fluorometholone and 0.05% Azelastine in Eyes with Severe Allergic Conjunctival Diseases: A randomized controlled trial”, which are helpful to the revision of our manuscript, and it definitely improve the quality of our manuscript. In this article, we have revised the manuscript in accordance with the reviewers’ comments and carefully proofread the manuscript to correct typographical, grammatical, and bibliographical errors. The revised texts are in red. In addition, our responses to the reviewers’ comments one by one are listed below. We hope the revised manuscript will be more acceptable for publication in your journal.

Best regards

Yours Sincerely,

Jiaxu Hong

Jiaxu Hong MD, PhD, MPA

Email: jiaxu_hong@163.com

Address: Department of Ophthalmology and Visual Science, Eye, and ENT Hospital, Shanghai Medical College, Fudan University, 83 Fenyang Road, Shanghai, China; Department of Ophthalmology, The Affiliated Hospital of Guizhou Medical University, Guiyang, China, Telephone: +86-021-64377134 Fax: +86-021-64377151

 It was good to include data at 6 weeks as the mast cell stabilizing effects of azelastine take time to work. It would be good to have even longer data, utilizing the effects of azelastine.

Response: Yes, indeed. The treatment course of 6 weeks is far from sufficient for severe ACD. This was a limitation discussed in our paper. However, the combination usage of fluorometholone and azelastine showed its priority than the fluorometholone alone since 1 week after medication in both signs and symptoms. So we consider the dual-acting azelastine can act fast as acknowledged in previous paper. Future data should be provided to verify the longer effects of azelastine as well as its safety.

- The conclusion makes sense as using azelastine in the compound takes advantage of the anti-histamine effect as well as the immunomodulatory effect of FML. 

Response: Yes, it is. We want to provide quick relief as soon as possible with the combined medication.

- You could comment on FML's poor ocular penetration thus limiting steroid IOP response and keeping its effect to the corneal surface.

Response: We did it in the revised paper.

Reviewer 3 Report

The topic of this study is timely and will of interest to the readers of the journal. Some images and tables were utilized in order to make the content of the text clearer. Τhe statistical analysis used was appropriate. However, some corrections are suggested to be made.

Corrections of major imprtance

-Power analysis should be performed to determine the adequacy of the sample.

-Τhe authors  should specify the rating scales used to evaluate the corneal involvement score, the palpebral conjunctiva papillae score and the conjunctival hyperemia score.

-The authors should clarify in the discussion section if there are corresponding studies with the present one and compare the results.

Corrections with minor importance

-Please indicate the number of children who participated in the study.

Author Response

Dear reviewer:

We really appreciate for the your comments on our manuscript titled “Combination Therapy of 0.1% Fluorometholone and 0.05% Azelastine in Eyes with Severe Allergic Conjunctival Diseases: A randomized controlled trial”, which are helpful to the revision of our manuscript, and it definitely improve the quality of our manuscript. In this article, we have revised the manuscript in accordance with the reviewers’ comments and carefully proofread the manuscript to correct typographical, grammatical, and bibliographical errors. The revised texts are in red. In addition, our responses to the reviewers’ comments one by one are listed below. We hope the revised manuscript will be more acceptable for publication in your journal.

Best regards

Yours Sincerely,

Jiaxu Hong

Jiaxu Hong MD, PhD, MPA

Email: jiaxu_hong@163.com

Address: Department of Ophthalmology and Visual Science, Eye, and ENT Hospital, Shanghai Medical College, Fudan University, 83 Fenyang Road, Shanghai, China; Department of Ophthalmology, The Affiliated Hospital of Guizhou Medical University, Guiyang, China, Telephone: +86-021-64377134 Fax: +86-021-64377151

The topic of this study is timely and will of interest to the readers of the journal. Some images and tables were utilized in order to make the content of the text clearer. Τhe statistical analysis used was appropriate. However, some corrections are suggested to be made.

Corrections of major imprtance

-Power analysis should be performed to determine the adequacy of the sample.

Response: We used the Two sample t-tests allowing unequal variance method to determine the adequacy of the sample. We added it in the methods part in the revised paper.

-Τhe authors  should specify the rating scales used to evaluate the corneal involvement score, the palpebral conjunctiva papillae score and the conjunctival hyperemia score.

Response: We added table 1 in the revised paper.

-The authors should clarify in the discussion section if there are corresponding studies with the present one and compare the results.

Response: We added it in the first paragraph of discussion part.

Corrections with minor importance

-Please indicate the number of children who participated in the study.

Response: We did it in the results part in the revised paper.
